# Uterine Carcinosarcoma (UCS): A Literature Review and Survival Analysis from a Retrospective Cohort Study

**DOI:** 10.3390/cancers16233905

**Published:** 2024-11-21

**Authors:** Mauro Francesco Pio Maiorano, Gennaro Cormio, Brigida Anna Maiorano, Vera Loizzi

**Affiliations:** 1Unit of Obstetrics and Gynecology, Department of Interdisciplinary Medicine (DIM), University of Bari “Aldo Moro”, Policlinico of Bari, Piazza Giulio Cesare 11, 70124 Bari, Italy; n.cormio@gynecology2.uniba.it (G.C.); vera.loizzi@uniba.it (V.L.); 2Gynecologic Oncology Unit, IRCCS Istituto Tumori “Giovanni Paolo II”, 70124 Bari, Italy; 3Department of Medical Oncology, IRCCS San Raffaele Hospital, Via Olgettina 60, 20132 Milan, Italy; maiorano.brigida@hsr.it

**Keywords:** uterine carcinosarcoma, uterine carcinosarcoma prognostic factors, retrospective cohort study, uterine carcinosarcoma overall survival, uterine carcinosarcoma lymphadenectomy, uterine carcinosarcoma staging, uterine carcinosarcoma therapeutic strategies, malignant mixed Müllerian tumour, MMMT uterus, rare gynaecological cancers

## Abstract

Uterine carcinosarcoma is a rare and aggressive cancer. This study aims to better understand the characteristics and outcomes of patients with this type of cancer by reviewing the medical records of 80 individuals treated over several years. We analysed the factors that impact on patient survival, including age, tumour size, and stage of cancer at diagnosis. Our findings emphasize the importance of early detection and tailored treatment strategies to improve UCS patients’ outcomes. This research may help guide future treatment approaches for this challenging disease and provide valuable insights for healthcare professionals dealing with similar cases.

## 1. Introduction

### 1.1. Epidemiology and Diagnosis of Uterine Carcinosarcoma

Uterine carcinosarcomas (UCSs), or endometrial carcinosarcomas, historically referred to as malignant mixed Müllerian tumours (MMMT), are rare and aggressive cancers that typically develop in the uterus. Characterized by both carcinomatous and sarcomatous components, UCS is believed to have a monoclonal epithelial origin [1]. Epidemiological data are limited due to the rarity and complex diagnosis of UCS, which constitutes approximately 2–5% of all uterine malignancies and has an estimated incidence of 2–5 per 100,000 women annually in the United States [2,3]. UCS has a poor prognosis, with 5-year survival rates between 18% and 47%, and accounts for 16.4% of deaths from uterine cancers, despite its low incidence [4,5]. Risk factors include advanced age, particularly in postmenopausal women, pelvic radiation exposure, and genetic predispositions, including TP53 mutations and BRCA mutations, especially in cases with ovarian primitivity [3,4,6,7]. UCS is commonly associated with TP53 mutations (60–97%) in serous-like phenotypes and lower prevalence of mutations like ARID1A, KRAS, PTEN, and PIK3CA in endometrioid-like subtypes [8,9,10,11]. In utero exposure to DES, as well as obesity and nulliparity, may also increase UCS risk [1,5,12,13,14]. Clinically, UCS often presents with non-specific symptoms, such as postmenopausal bleeding or pain, making it difficult to distinguish from other endometrial cancers [15]. Diagnostic confirmation generally requires endometrial biopsy, although definitive diagnosis may be made post-hysterectomy. Staging tools include transvaginal ultrasound, MRI, and CT or PET scans, with CA125 levels aiding in prognosis and follow-up [16,17,18]. UCS spreads predominantly via lymphatic and intraperitoneal routes, more like epithelial tumours, contributing to its aggressive clinical course. Stage progression is closely linked to survival rates, from ~55% for Stage I to ~10% for Stage IV [9,19,20,21].

### 1.2. Pathological and Molecular Features of Uterine Carcinosarcoma

Initially classified as a high-grade uterine sarcoma, UCS is now recognized as a high-risk endometrial carcinoma with epithelial de-differentiation or metaplasia [19,21]. The World Health Organization’s 2020 classification included UCS as a high-grade carcinoma, requiring identification of the epithelial component for accurate diagnosis [22]. UCS comprises biphasic malignancies with adenocarcinomatous and sarcomatous elements, with the sarcomatous component showing homologous or heterologous differentiation. Sarcomatous predominance or heterologous differentiation, such as rhabdomyosarcoma, correlates with poorer prognosis [10,23]. Molecularly, UCS may originate from a single epithelial clone with sarcomatous transformation, supported by TCGA’s four molecular subgroups within endometrial cancer [24,25]. Limited data on UCS subtypes indicate a majority align with the p53-abnormal group, associated with aggressive behaviour and poor prognosis. UCS cases with POLE mutations, however, align with low-risk profiles [9,21,26,27,28]. Further research is needed to establish the significance of MSI-H subgroups in UCS outcomes [29,30,31,32].

### 1.3. Uterine Carcinosarcoma Therapeutic Strategies and Aim of the Study

Given the rarity of UCS, the evidence regarding its standard of care is limited, mostly derived from retrospective or non-randomized studies, often with small populations. Currently, no definitive consensus exists on its optimal management. However, aligning with ESGO/ESTRO/ESP recommendations and National Comprehensive Cancer Network (NCCN) guidelines, treatment approaches for UCS mirror those for other non-endometrioid high-grade endometrial cancers [28,33]. The primary treatment approach is a multimodal strategy involving surgery, chemotherapy, and/or radiotherapy. Standard surgical staging for UCS includes hysterectomy, bilateral salpingo-oophorectomy, omentectomy, peritoneal biopsies, and lymphadenectomy for patients in early stages. Minimally invasive surgery is preferred for apparent early-stage disease, whereas open abdominal cytoreductive surgery may be suitable for advanced cases. Ovarian preservation and fertility-sparing procedures are generally contraindicated in UCS. Even for Stage I carcinosarcomas, including serous and undifferentiated subtypes, infra-colic omentectomy and random peritoneal biopsies are part of the staging process, although data on the role of peritoneal staging in cases with endometrioid or other non-endometrioid components remain limited [28,33]. For lymph node assessment, resection of visibly enlarged nodes is recommended in advanced stages, but the therapeutic value of systematic lymphadenectomy in early-stage UCS remains uncertain. Sentinel node mapping has emerged as a viable alternative for nodal staging in UCS. Studies on high-risk endometrial cancers suggest its accuracy and safety, with a false-negative rate below 1% [33,34]. Specifically for UCS, research indicates no significant difference in progression-free survival (PFS) between patients undergoing sentinel node mapping and those having standard lymphadenectomy [35]. A recent retrospective study comparing UCS patients undergoing sentinel node mapping versus systematic lymphadenectomy demonstrated that sentinel mapping effectively detects nodal metastasis without affecting overall oncologic outcomes [36]. The FIRES trial further supports the diagnostic accuracy of sentinel node mapping using indocyanine green for detecting nodal metastases, suggesting its potential as a substitute for full lymphadenectomy across various endometrial cancer types [37]. Similarly, the multicentre SENTOR trial found that indocyanine green sentinel node mapping in high-risk endometrial cancers, including UCS, provided diagnostic accuracy comparable to or exceeding that of systematic lymphadenectomy [38]. However, UCS cases were limited in these studies, prompting ongoing trials, such as SNEC [39], ALICE [40], ENDO-3 [41], and ECLAT [42], to further investigate nodal staging in high-risk endometrial cancers. Presently, there is no consensus on the need for lymphadenectomy in early-stage UCS patients who have already undergone hysterectomy with no nodal staging and negative imaging results. In such cases, additional surgery may be avoided in favour of adjuvant radiotherapy and chemotherapy to address nodal risk areas. Based on traditional and molecular classifications, adjuvant therapy typically involves chemotherapy and radiotherapy, with considerations for the molecular profile and risk stratification. Chemotherapy, particularly the carboplatin/paclitaxel regimen, plays a central role in treatment, with concurrent chemoradiation showing promise in reducing recurrence risk and improving survival rates [28,33,43]. Immunotherapy, particularly pembrolizumab and lenvatinib, is emerging as a standard treatment option following platinum-based chemotherapy failure, although further validation is needed, especially for UCS [44]. For recurrent disease, second-line therapy options include platinum-based chemotherapy rechallenge and various other agents, while the role of radiotherapy is more limited, typically in combination with chemotherapy. In the context of metastatic disease, endocrine therapy may be considered as an alternative treatment option particularly for hormone-receptor-positive tumours, albeit with limited data supporting its efficacy in UCS [28,33]. As clear guidelines regarding UCS management are lacking, we performed a retrospective observational cohort study aiming to assess the clinical and pathological features as well as prognostic biomarkers of UCS through a comprehensive analysis of 80 patients, alongside a detailed review of the existing literature. The study aims to identify key predictors of survival and explore their potential associations, ultimately contributing to improving management strategies for this rare and aggressive malignancy.

## 2. Materials and Methods

Data on all cases of uterine carcinosarcoma assessed between 1995 and 2024 at three Italian hospitals (IRCCS Istituto Oncologico “Giovanni Paolo II”, Bari, Italy; Azienda Ospedaliero Universitaria Policlinico di Bari, Bari, Italy; IRCCS “Casa Sollievo della Sofferenza”, San Giovanni Rotondo, Italy) were collected. We included patients with histologically confirmed uterine carcinosarcoma (UCS) at any stage of the disease, who received both primary treatment and follow-up care at one of the three participating hospitals. Eligible patients had to be aged 18 years or older, as the study focused on adult populations typically affected by UCS. Only patients who had comprehensive medical records with information relevant to the study’s objectives, such as clinical symptoms, diagnostic and surgical details, tumour characteristics, and follow-up data, were considered in this study. Patients were excluded if they had incomplete medical records or insufficient follow-up data, as these could limit accurate assessments of survival outcomes. For ethical reasons, patients under 18, pregnant women, and those with psychological conditions that would preclude informed consent were excluded. Additionally, patients with significant comorbidities that could confound survival data, such as other advanced malignancies, were also excluded to minimize bias in interpreting UCS-specific prognostic outcomes. Information was extracted from medical records and follow-up charts, encompassing details such as age at diagnosis, parity, menopausal status, family or personal cancer history, clinical symptoms, diagnosis, tumour markers (CA125, CA15.3, and CA19.9), hysteroscopic findings, primary treatment, type of surgery, residual disease, staging (based on the FIGO 2009 system for endometrial carcinoma [45]; restaging was carried out accordingly if previous or different staging systems were used), histological features, metastases, adjuvant chemotherapy (cycle count, types, and dosages), disease recurrence, disease-free interval, radiation therapy specifics, and patient status at the last follow-up. Tumour histology was assessed to include the maximum diameter (tumour size) and the presence and proportion of sarcomatous and carcinomatous components, noting whether the sarcomatous component was homologous or heterologous and if sarcomatous dominance (greater than 50% of the tumour) was observed. The specific differentiation patterns (e.g., rhabdomyosarcoma and chondrosarcoma) within the sarcomatous component, as well as high-grade serous, endometrioid, or undifferentiated forms within the epithelial component, were documented. Incomplete data were taken into consideration, particularly for patients with brief follow-up durations. To ensure data reliability, patients with follow-up periods under six months were excluded unless death occurred. Efforts to mitigate biases from incomplete data included a review of all records by two independent researchers to confirm eligibility, with disagreements resolved through consensus. Anonymized patients’ data are available (Appendix A). Statistical analyses were performed using SPSS software version 24 [46]. Survival analyses were conducted using the Kaplan–Meier method to estimate overall survival (OS). Survival curves were compared across groups using the log-rank test to evaluate differences in survival distributions [47]. Cox proportional hazards regression models were used to identify independent prognostic factors in both univariate and multivariate analyses [48]. Variables with a *p*-value <0.05 in the univariate analysis were included in the multivariate model using both the enter method and a backward stepwise approach. In the Cox regression analyses, age was treated as a continuous variable, while tumour size was categorized into two groups, ≤4 cm and >4 cm, based on the maximum diameter of the tumour. Tumour stage (I–II vs. III–IV), myometrial invasion (≤50% and >50%), histotype (homologous and heterologous), and lymphadenectomy (performed/not performed) were considered categorical variables in the analyses, as specified in Section 3.2. Hazard ratios (HRs) with 95% confidence intervals (CIs) were reported. A *p*-value < 0.05 was considered statistically significant.

## 3. Results

### 3.1. Characteristics of the Included Patients

Eighty patients diagnosed with uterine carcinosarcoma were included in this study, with a median age of 66 years (range: 42–86 years). Table 1 presents the primary clinical characteristics.

Among these cases, metrorrhagia was the initial symptom in 31 patients (38.75%), and 18 patients (22.5%) reported vaginal spotting. Seven patients (8.75%) were asymptomatic, and in two cases (2.5%), the diagnosis was incidental (one via ultrasound and one via CT scan). Abdominal pain was frequent, affecting 19 patients (23.75%), and 4 patients (5%) had a palpable abdominal mass. Serum CA-125 was positive in 15 patients (18.75%) and negative in 31 (38.75%); for the remaining 34 cases (42.5%), CA-125 data were unavailable. Elevated CA-15.3 and CA-19.9 levels were observed in 5 patients (6.25%), while values were within the normal range for 50% of cases. In terms of performance status, 49 patients (61.25%) had an ECOG (Eastern Cooperative Oncology Group) score of 0, two patients had scores of 1 and 2, respectively, with ECOG scores unavailable for the remaining cases. Hysteroscopy findings were polypoid in 57% and diffusely irregular in 41% of cases. Surgical intervention, primarily via total abdominal hysterectomy with bilateral salpingo-oophorectomy (TAH-BSO), was performed for all 80 patients. Three underwent laparoscopic surgery, and one had robotic surgery. Omentectomy was completed in 18 patients (22.5%), with 2 (2.5%) undergoing omental biopsy; peritoneal washing was performed in 15 cases (18.75%). Pelvic lymphadenectomy was performed in 42 cases (52.5%), and lymph node sampling was carried out in 4 cases (5%). Histology revealed heterologous carcinosarcoma in 48 patients (60%) and homologous type in 28 (35%); histologic type was not specified for 10 patients (12.5%). Myometrial invasion was ≤50% in 43 cases (53.75%) and >50% in 25 cases (31.25%), with the extent undetermined in the remaining 12 cases. Tumour size ranged from 1 to 20 cm, and 10 of the 42 patients (23.81%) who underwent lymphadenectomy had positive nodes; no positive nodes were found in the four sampled cases. Staging by FIGO showed that 34 patients (42.5%) were Stage I, with specific subtypes IA, IB, and IC assigned in 11, 12, and 7 cases, respectively, while 4 cases were unspecified (NS). Stage II was found in 11 patients (13.75%), of which 6 were NS, 4 were IIB, and 1 was IIC. Twenty patients (25%) were Stage III, with further subdivisions IIIA, IIIB, and IIIC observed in 4, 3, and 10 cases, respectively. Fifteen patients (18.75%) had Stage IV disease, with metastases commonly observed in the omentum, peritoneum, and lymph nodes. Adjuvant chemotherapy was given to 54 patients (67.5%), predominantly carboplatin and paclitaxel (CBDCA + TAX) in 39 patients (48.75%), with an average of 5.5 cycles; 14 patients (17.5%) received epirubicin and ifosfamide, averaging 4.5 cycles. Treatment response was assessed in 51 of the 54 patients (94.44%), primarily through CT scans, resulting in 34 complete responses (66.67%), 5 partial responses (9.80%), 4 stable diseases (7.84%), and 8 cases of disease progression (15.69%). Two patients also received chemoradiation therapy. Recurrence occurred in 26 cases (32.5%), with a disease-free interval (DFI) ranging from 1 to 143 months (median DFI: 17.92 months; 95% CI: 8.09–27.75 months). Common recurrence sites included the lymph nodes, peritoneum, and above the vaginal cupola, with isolated liver and lung metastases in some cases. Recurrence diagnosis was primarily confirmed by CT (11 cases), with MRI, PET scan, and vaginal biopsy used in other cases. Nineteen recurrences were at a single site, while seven involved multiple sites, with tumour size at relapse ranging from 2 to 48 cm. Elevated CA-125 levels were observed in four relapsed cases, with concurrent CA-15.3 elevation in two cases. Nine patients received second-line chemotherapy, primarily doxorubicin, epirubicin, CBDCA+TAX rechallenge, or cyclophosphamide, showing limited response (six cases of progression and three cases of stable disease). Eight patients underwent surgical resection of recurrent tumours, followed by chemotherapy in three cases and pelvic radiotherapy in two cases, resulting in three cases of stable disease and three complete responses.

### 3.2. Survival Outcomes and Univariate Analyses for OS

At the last follow-up, 3 patients (3.75%) were alive with disease (AWD), 44 (55%) were dead of disease (DOD), 2 (2.5%) died of other causes (DOC), and 31 (38.75%) were alive with no evidence of disease (NED). The median overall survival (mOS) of our cohort was 34.5 months (95% CI, 23.5–50.0 months). We ran Cox proportional hazards regression for univariate analyses and identified tumour stage, tumour size, and age as significant predictors of survival. Tumour stage was significantly associated with survival (*p* = 0.008). Patients with Stage I–II tumours had a 56.2% lower risk of experiencing the event (death) compared to those with Stage III–IV tumours (HR = 0.438; 95% CI: 0.238–0.809), indicating that advancing tumour stage is strongly correlated with decreased survival. Similarly, tumour size was a significant predictor of survival (*p* = 0.019). Patients with tumours larger than 4 cm were at more than twice the risk of death as those with tumours that were 4 cm or smaller (HR = 2.154; 95% CI: 1.135–4.087), suggesting that larger tumour size is associated with poorer survival outcomes. Lastly, age also emerged as a significant factor in predicting survival (*p* = 0.002). For each additional year of age, the risk of experiencing the event increased by 6.4% (HR = 1.064; 95% CI: 1.024–1.105), demonstrating that increasing age is independently linked to a higher risk of death. Other factors, such as histotype, myometrial invasion, and lymphadenectomy, did not show significant associations with survival outcomes at the univariate analyses (*p* > 0.05, Table 2).

### 3.3. Multivariate Analyses for OS

A multivariate Cox proportional hazards regression was conducted to assess the independent effects of these variables. The multivariate analyses revealed that both age and tumour stage were independent predictors of survival. For each additional year of age, the risk of death increased by 6.7% (HR = 1.067; 95% CI: 1.024–1.111; *p* = 0.002). Tumour stage was also significantly associated with survival, with patients having Stage III–IV tumours showing a 52% increased risk of death compared to those with Stage I–II tumours (HR = 0.483; 95% CI: 0.257–0.906; *p* = 0.023). Tumour size, however, was not a statistically significant predictor in the multivariate model (*p* = 0.206) (Table 3).

Given the results of the initial multivariate analysis, a backward stepwise multivariate Cox proportional hazards regression was conducted to further refine the model, explore potential associations with both significant and non-significant variables, and thus identify independent predictors of survival. The final model retained age and tumour stage as significant predictors. For each additional year of age, the hazard of death increased by 4.6% (HR = 1.046; 95% CI: 1.005–1.089; *p* = 0.027). Tumour stage was also a significant predictor, with patients in Stage III–IV having a 53.4% higher risk of death compared to those in Stage I–II (HR = 0.466; 95% CI: 0.241–0.902; *p* = 0.023). Other covariates, including tumour size, myometrial invasion, histotype, and lymphadenectomy, were not significant and were excluded from the final model (Table 4).

### 3.4. Kaplan–Meier Survival Analyses

Following the identification of significant predictors through univariate and multivariate Cox proportional hazards regression analyses, Kaplan–Meier survival curves were generated to further explore survival differences across key variables (FIGO stage and histological definition of uterine carcinosarcomas, tumour size, pelvic lymphadenectomy, and myometrial invasion). This approach allowed for the visual representation of survival probabilities over time and provided additional insights into potential associations, particularly for variables that did not reach significance in the Cox models. Kaplan–Meier survival analysis by tumour stage revealed a statistically significant difference in survival between patients with early-stage (Stage I–II) and advanced-stage (Stage III–IV) uterine carcinosarcoma (log-rank *p* = 0.027) (Figure 1).

For histotype, Kaplan–Meier survival analysis did not reveal a statistically significant difference (*p* = 0.470) between homologous and heterologous types of uterine carcinosarcoma (Figure 2).

When histotype was stratified by tumour stage, Kaplan–Meier analysis revealed a borderline significant difference in survival between patients with homologous and heterologous histotypes (log-rank *p* = 0.050). Specifically, survival outcomes appeared to differ more clearly within advanced tumour stages (III–IV) compared to earlier stages (I–II). In the multivariate Cox proportional hazards regression, histotype did not emerge as a significant independent predictor of survival, suggesting that the effect of histotype on survival is likely influenced by other factors, such as tumour stage, which was a significant predictor in the Cox analysis (Figure 3).

Tumour size also showed a significant association with survival. Kaplan–Meier analyses comparing tumour size categories without stratification by tumour stage revealed a significant difference in survival between patients with tumours larger than 4 cm and those with tumours 4 cm or smaller (log-rank *p* = 0.015). Patients with tumours > 4 cm had a median survival of 35 months, compared to 142 months for those with tumours ≤ 4 cm, indicating a significant association between tumour size and survival. While tumour size did not remain significant in the multivariate Cox proportional hazards regression, probably influenced by tumour stage, these Kaplan–Meier results emphasize the importance of tumour size as a predictor of survival: its effect is more pronounced when considered alongside tumour stage (Figure 4).

Further Kaplan–Meier analysis stratified by tumour stage revealed a highly significant difference in survival based on tumour size (log-rank *p* = 0.003), with patients having tumours > 4 cm experiencing markedly poorer survival, particularly in Stage III–IV disease, where their median survival was only 26 months, compared to those with tumours ≤ 4 cm, whose median survival was not reached. Although tumour size was not significant in the multivariate Cox analysis, the Kaplan–Meier results suggest that tumour size remains a crucial predictor of survival, especially when considered in the context of tumour stage (Figure 5).

For myometrial invasion alone, Kaplan–Meier survival analysis did not reveal a significant difference in survival between patients with ≤50% invasion and those with >50% invasion (log-rank *p* = 0.276). However, when stratified by tumour stage, a significant difference in survival was observed (log-rank *p* = 0.002). Patients with Stage III–IV tumours and myometrial invasion >50% had a median survival of 21 months, compared to Stage I–II patients, where median survival was not reached. These findings suggest that myometrial invasion influences survival primarily in patients with advanced-stage disease (Figure 6 and Figure 7).

Our Kaplan–Meier analysis revealed a significant association between lymphadenectomy and improved survival outcomes (log-rank *p* = 0.037), with a median survival of 67 months for patients who underwent lymphadenectomy compared to 51 months for those who did not (Figure 8).

Further Kaplan–Meier survival analysis stratified by tumour stage confirmed a statistically significant difference in survival between patients who underwent lymphadenectomy and those who did not (*p* = 0.039) (Figure 9). 

Finally, further analysis within Stage I–II and Stage III–IV subgroups showed no significant survival difference based on lymphadenectomy (*p* = 0.736 and *p* = 0.724, respectively). This suggests that while lymphadenectomy may be associated with survival when considering the overall population, it does not independently influence survival within specific tumour stages. These findings align with the results from the multivariate Cox regression, where lymphadenectomy was not a significant predictor of survival.

## 4. Discussion

The present retrospective cohort study on UCS, a rare and aggressive malignancy, provides a comprehensive analysis of the current UCS literature as well as of 80 patients treated at three institutions. The findings show an mOS of 34.5 months, with a recurrence rate of 32.5% and an mDFI of 17.92 months, reflecting the aggressive nature of UCS. These outcomes align with previous studies, where 5-year survival rates ranged from 18% to 47% depending on stage and treatment modality [49]. The overall poor prognosis underscores the need for better therapeutic strategies, as UCS continues to represent a clinical challenge with its propensity for recurrence and metastases. The Kaplan–Meier survival analysis stratified by FIGO stage revealed significant survival differences, with patients in Stage I–II demonstrating markedly better survival compared to those in Stage III–IV (log-rank *p* = 0.027). Specifically, the median survival was estimated at 101 months for Stage I–II, while it was only 30 months for Stage III–IV, emphasizing tumour stage as a pivotal determinant of survival in UCS. This finding is consistent with our multivariate Cox regression, which identified tumour stage as the most significant independent predictor of survival, where advanced stages (Stage III–IV) were associated with a substantially higher risk of mortality (HR = 0.438; *p* = 0.008). These observations reinforce established knowledge that early-stage diagnosis is associated with improved survival, underscoring the critical need for early detection strategies in UCS management [5,49].

We found a predominance of heterologous-type carcinosarcoma (60%) in our cohort, which is associated with poorer survival outcomes compared to the homologous type (35%). Heterologous sarcomatous differentiation, especially with elements like rhabdomyosarcoma or chondrosarcoma, is known to correlate with more aggressive behaviour and a worse prognosis. Findings from the literature highlighted similar outcomes, with heterologous differentiation being a poor prognostic indicator in UCS [50]. Adjuvant chemotherapy, mainly carboplatin and paclitaxel, was administered to 67.5% of patients in this cohort, achieving a complete response in 66.67% of cases. This response rate is consistent with other studies, which have established carboplatin and paclitaxel as the cornerstone regimen for UCS [43,51,52]. In addition, the use of doxorubicin-based regimens was less common but observed in 17.5% of cases, echoing its historical usage in UCS treatment, though its efficacy has been debated due to its increased toxicity profile and lower response rates [52,53,54]. The recurrence rate of 32.5% and the predominance of local relapses (19 out of 26 cases) are significant, reflecting UCS’s aggressive nature and the challenges in achieving long-term disease control. Studies support the finding that UCS metastasizes early, primarily to lymph nodes and the peritoneum, necessitating aggressive multimodal treatment [55]. This study’s recurrence pattern, combined with high CA-125 levels observed in some relapsed cases, aligns with prior research suggesting that CA-125 is a marker for poor prognosis and recurrence in UCS [56,57,58]. The results of our retrospective cohort study of 80 patients with UCS largely align with previously reported findings in the literature, while also contributing additional insights into the prognostic factors associated with survival in this rare and aggressive malignancy. Our study identified tumour stage as the most significant independent predictor of survival, consistent with numerous prior studies that emphasize the pivotal role of the stage in UCS prognosis. Specifically, patients with Stage I–II tumours demonstrated a significantly lower risk of death compared to those with Stage III–IV tumours (HR = 0.438; *p* = 0.008), reinforcing the established understanding that early-stage diagnosis is associated with improved survival. This aligns with findings from Nemani et al., who also reported stage as the strongest determinant of survival, with advanced-stage disease correlating with significantly poorer outcomes [5,49,59]. Our analyses also identified age as an independent predictor of survival in both univariate and multivariate Cox proportional hazards regression analyses. For each additional year of age, the risk of death increased by 6.4% (HR = 1.064; *p* = 0.002), confirming that age plays a significant role in survival outcomes for UCS. This finding aligns with previous research, where older age has been consistently associated with poorer prognosis in UCS and other high-grade endometrial cancers [43,49,60]. The higher mortality in older patients may be related to a combination of factors, including increased comorbidities, decreased physiological reserve, and potentially delayed diagnosis. Furthermore, the diminished capacity to tolerate aggressive multimodal treatments, which are typically employed for UCS, might contribute to the worse outcomes observed in older patients. Future treatment protocols should consider age as a key factor when tailoring individualized treatment plans, balancing the potential benefits of aggressive therapy with the risks of morbidity in older patients. Tumour size emerged as another significant predictor of survival in our univariate analysis, with patients harbouring tumours larger than 4 cm facing more than twice the risk of death compared to those with smaller tumours (HR = 2.154; *p* = 0.019). Although tumour size did not retain its significance in the multivariate model, likely due to the confounding influence of advanced-stage disease, its significant impact in the Kaplan–Meier analysis stratified by stage (log-rank *p* = 0.003) underscores its importance, particularly in advanced disease. Larger tumour size has been previously associated with poorer outcomes in UCS and other high-risk endometrial cancers. Our findings are in line with those from the literature, which reported that larger tumour size was a poor prognostic factor, particularly when combined with high-grade histopathological features [5]. The reduced significance of tumour size in multivariate analysis suggests that advanced stage may subsume the impact of tumour size on survival, as larger tumours are often associated with later-stage disease. This pattern indicates that stage may be a more comprehensive marker of disease progression in UCS. Our investigation into histotype revealed no significant difference in survival between patients with homologous and heterologous histotypes in both the Cox regression and Kaplan–Meier analyses (log-rank *p* = 0.050). These results suggest that while histotype is an important pathological characteristic, its independent effect on survival is limited once other factors, such as tumour stage, are considered. Prior research has similarly shown that histotype alone is not a robust predictor of survival, with studies indicating that histotype often interacts with other clinicopathological factors to influence outcomes [61,62]. Although myometrial invasion was not a significant independent predictor of survival in our multivariate analysis, Kaplan–Meier analysis stratified by stage indicated a significant association with survival in advanced-stage disease (log-rank *p* = 0.002). Specifically, patients with Stage III–IV tumours and >50% myometrial invasion had a median survival of only 21 months. These findings are in agreement with the literature, which reported that deep myometrial invasion (>50%) was associated with poor prognosis, especially in advanced-stage UCS [63,64]. The stratification by stage in our analysis highlights the importance of considering the context in which invasion occurs, particularly in aggressive cancers like UCS, where stage likely amplifies the impact of invasion on survival outcomes. Interestingly, lymphadenectomy did not emerge as a significant independent predictor of survival in our study, echoing findings from other retrospective analyses. Despite initial Kaplan–Meier results indicating a significant difference in survival (log-rank *p* = 0.039), further analysis stratified by stage showed no survival difference between patients who underwent lymphadenectomy and those who did not (*p* = 0.736 for Stage I–II and *p* = 0.724 for Stage III–IV). This aligns with the findings from the literature, which found no improvement in progression-free survival among UCS patients undergoing lymphadenectomy, raising questions about its therapeutic benefit in this population, while underlining the importance of sentinel lymph node mapping. Our results suggest that while lymphadenectomy may have a role in staging and diagnosis, it is unlikely to significantly impact long-term survival, particularly when controlling for tumour stage and other factors [35]. This could partially depend on the presence of the sarcomatous component, which instead typically spreads through the haematological way. Overall, our findings highlight the critical role of tumour stage and age as independent predictors of survival in UCS, while underscoring the context-dependent nature of other factors such as tumour size, myometrial invasion, and lymphadenectomy. These results are consistent with much of the existing literature and reinforce the need for a multimodal treatment approach that accounts for individual patient characteristics. Several limitations of this study must be acknowledged. First, as a retrospective cohort study, it is subject to selection bias, and the quality of data is dependent on the completeness and accuracy of medical records. Additionally, because UCS is a rare neoplasm, our relatively small sample size of 80 patients limits the statistical power of some analyses and may impact the generalizability of the results. Furthermore, while we performed multivariate analyses to control for confounding factors, residual confounding cannot be completely ruled out. Our study also lacked molecular profiling, which could have provided further insights into the underlying biology and potential therapeutic targets of UCS. Finally, our cohort is limited to patients from three centres in Italy, which may limit the generalizability of findings to populations with different genetic, environmental, or healthcare access factors. As UCS remains a rare and aggressive malignancy with limited prospective data, future studies should explore its molecular characterization more deeply, building on recent work from TCGA and similar groups, which have identified potentially prognostic molecular subtypes, such as POLE, MSI-H, and p53-abnormal [25]. With new molecular classifications and increased awareness of its pathogenesis, endometrial carcinosarcoma is gradually gaining attention in clinical trials after years of being overlooked. Numerous molecular studies have identified mutations or alterations in genes and pathways such as c-KIT, TKR, VEGF, EGFR, Her2/neu, NTRK, PI3K/AKT/mTOR pathway, WEE1, KRAS, EXP, BRCA1/2, and others related to cell-cycle regulation, histone modification, and chromatin remodelling, offering potential therapeutic targets [65,66]. Novel molecular-targeted therapies hold promise for the management of UCS, particularly in the setting of recurrence or metastasis, with ongoing prospective clinical trials investigating immune checkpoint inhibitors, HER2 targeting agents, and WEE1 inhibitors showing encouraging results [67,68,69]. The role of novel molecular-targeted therapies should also be prospectively explored in the context of these subtypes, especially in cases of recurrent or metastatic disease, where treatment options are currently limited. Additionally, future research should evaluate the efficacy of minimally invasive techniques, including sentinel node mapping, and assess the value of adjuvant chemoradiation and immunotherapy in improving survival outcomes. Further investigation is needed to determine the optimal treatment approach for elderly patients with UCS, balancing treatment aggressiveness with quality-of-life considerations [70].

## 5. Conclusions

Our observational retrospective cohort study gives valuable insights into the clinical and prognostic factors influencing UCS survival. Through comprehensive univariate and multivariate analyses, we identified tumour stage and age as the strongest independent predictors of survival, with advanced-stage disease and increasing age significantly correlating with poorer outcomes. These findings reaffirm the critical importance of early detection and timely intervention in improving survival rates for patients with UCS. This study also underscores the need for a multimodal treatment approach, integrating surgery, chemotherapy, and radiotherapy, particularly for patients with high-risk or advanced-stage disease. Even if chemotherapy remains the cornerstone of UCS therapy, novel therapeutic strategies should be explored, particularly for recurrent or metastatic cases where current treatment options remain limited. As we move towards the era of precision oncology, the integration of molecular diagnostics, multi-omics approaches, and tailored treatment strategies will be key to improving outcomes for patients facing this challenging malignancy. The development of machine learning may provide a chance to better integrate multidimensional data, ultimately leading to more effective and individualized care.

## Figures and Tables

**Figure 1 cancers-16-03905-f001:**
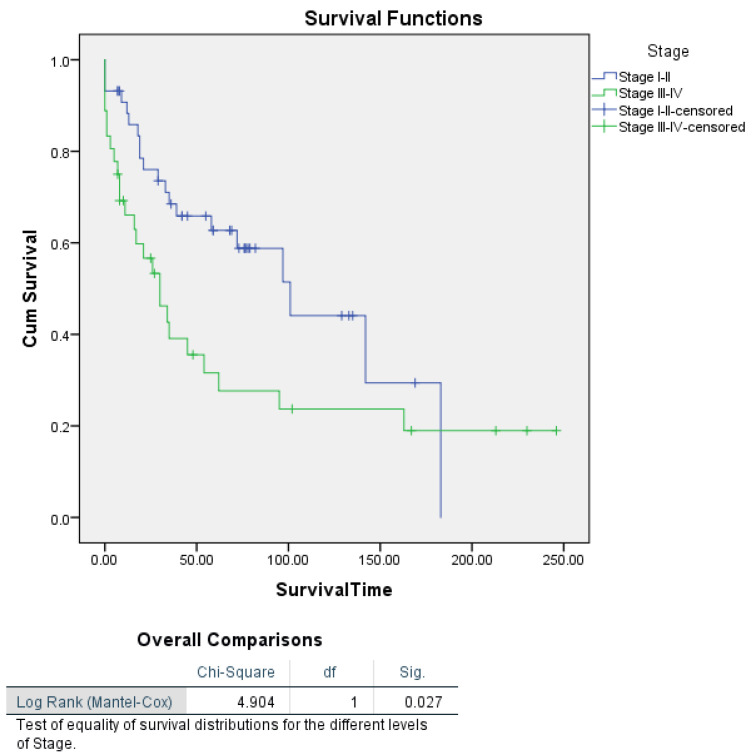
Kaplan–Meier survival curves for patients with uterine carcinosarcoma stratified by tumour stage (Stage I–II vs. Stage III–IV).

**Figure 2 cancers-16-03905-f002:**
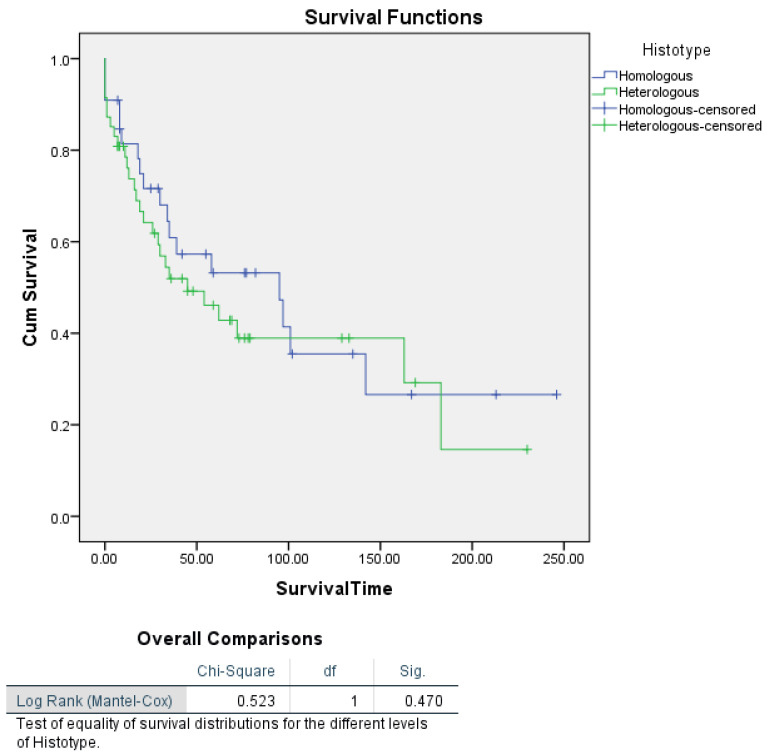
Kaplan–Meier analysis of disease-specific survival for patients with uterine carcinosarcoma by histotype.

**Figure 3 cancers-16-03905-f003:**
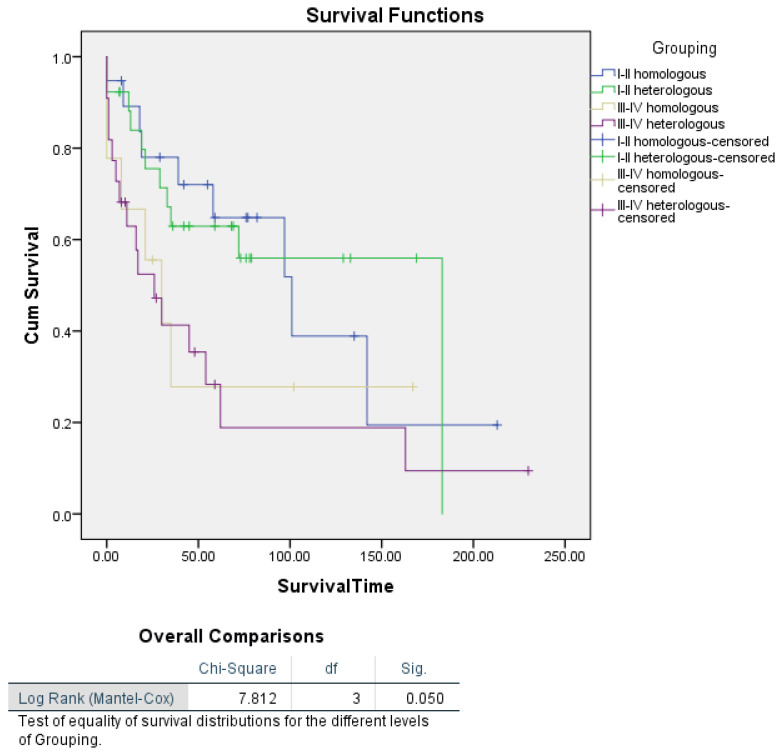
Kaplan–Meier analysis of disease-specific survival for patients with uterine carcinosarcoma by histotype, segmented by FIGO stage.

**Figure 4 cancers-16-03905-f004:**
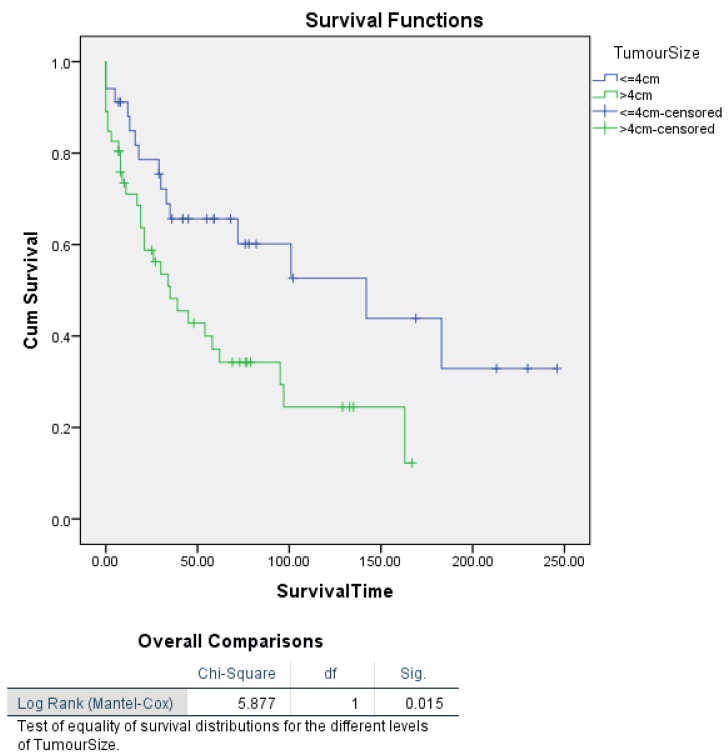
Kaplan–Meier survival analysis of disease-specific survival for uterine carcinosarcoma patients by tumour size, without FIGO stage segmentation.

**Figure 5 cancers-16-03905-f005:**
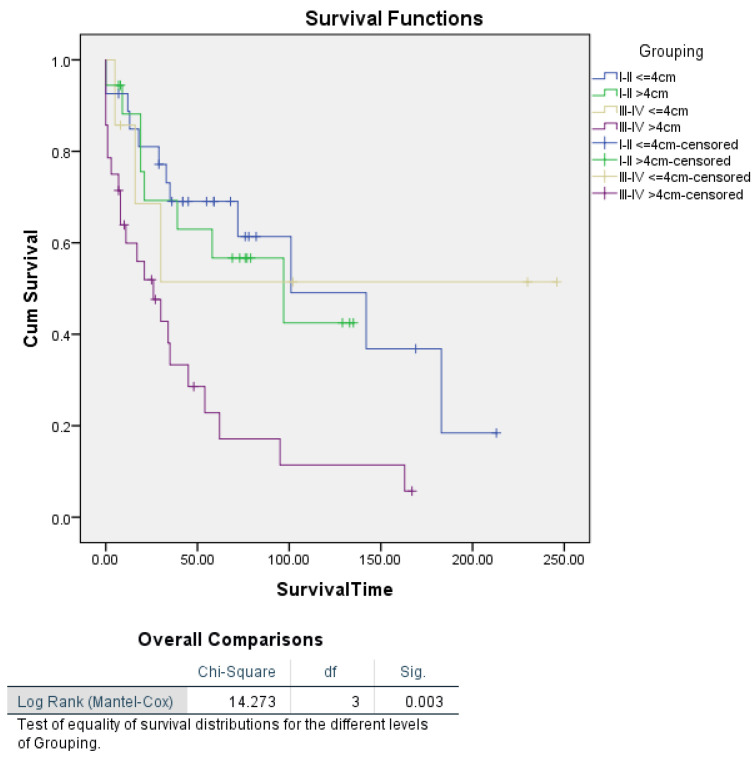
Kaplan–Meier survival curves illustrating disease-specific survival in uterine carcinosarcoma patients based on tumour size, segmented by FIGO stage.

**Figure 6 cancers-16-03905-f006:**
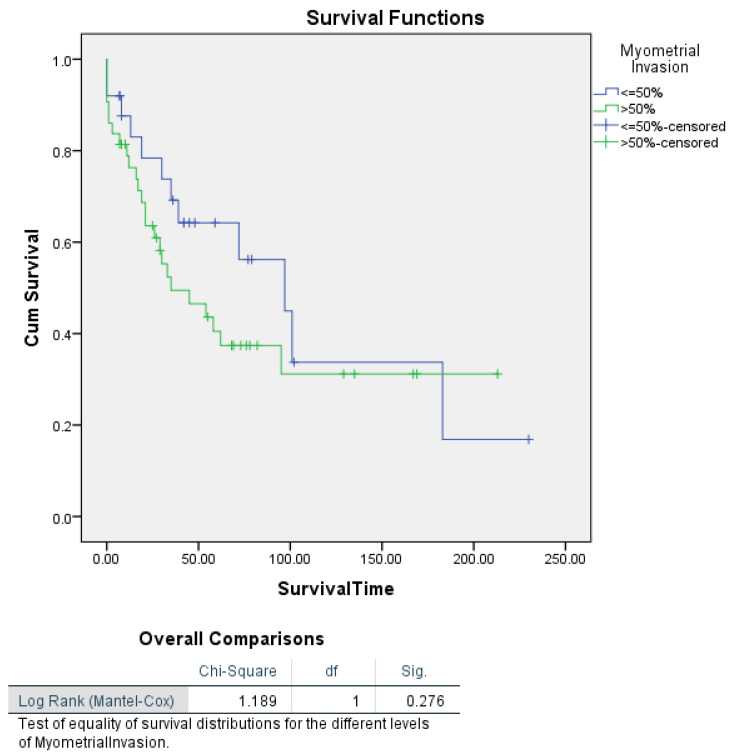
Kaplan–Meier disease-specific survival analysis for uterine carcinosarcoma patients according to myometrial invasion depth.

**Figure 7 cancers-16-03905-f007:**
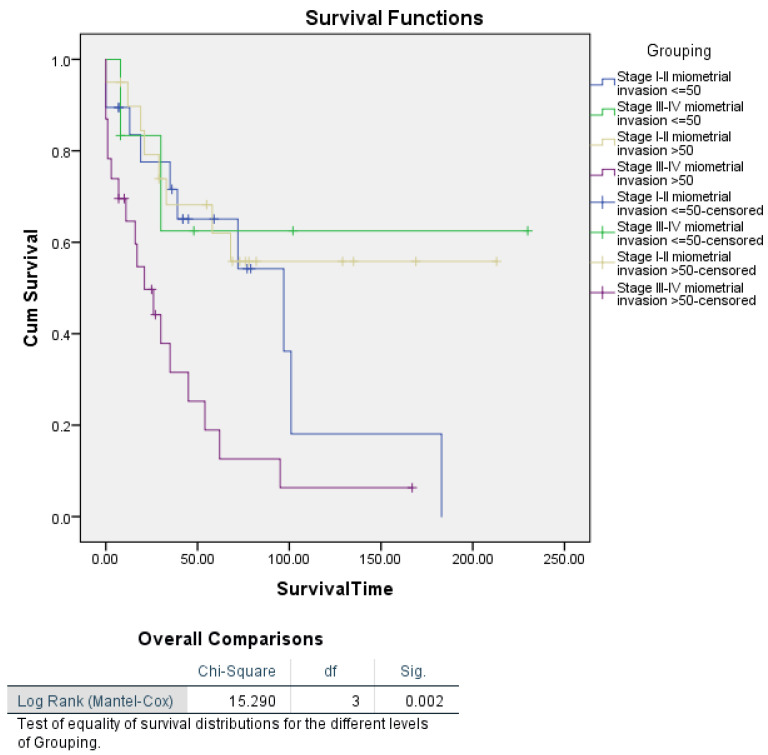
Kaplan–Meier curves showing disease-specific survival in uterine carcinosarcoma patients, categorized by myometrial invasion and segmented by FIGO stage.

**Figure 8 cancers-16-03905-f008:**
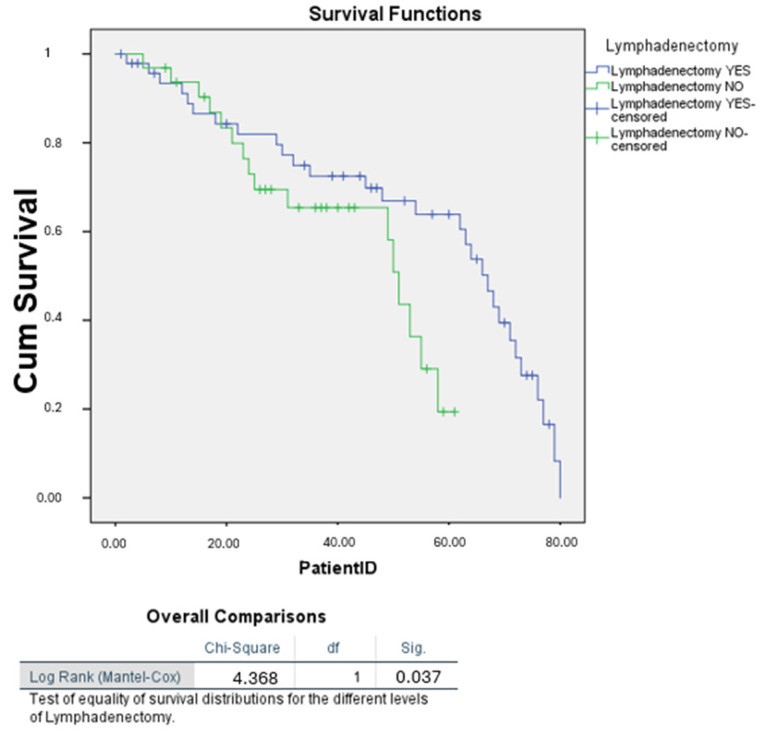
Kaplan–Meier survival curves for patients with uterine carcinosarcoma based on lymphadenectomy status.

**Figure 9 cancers-16-03905-f009:**
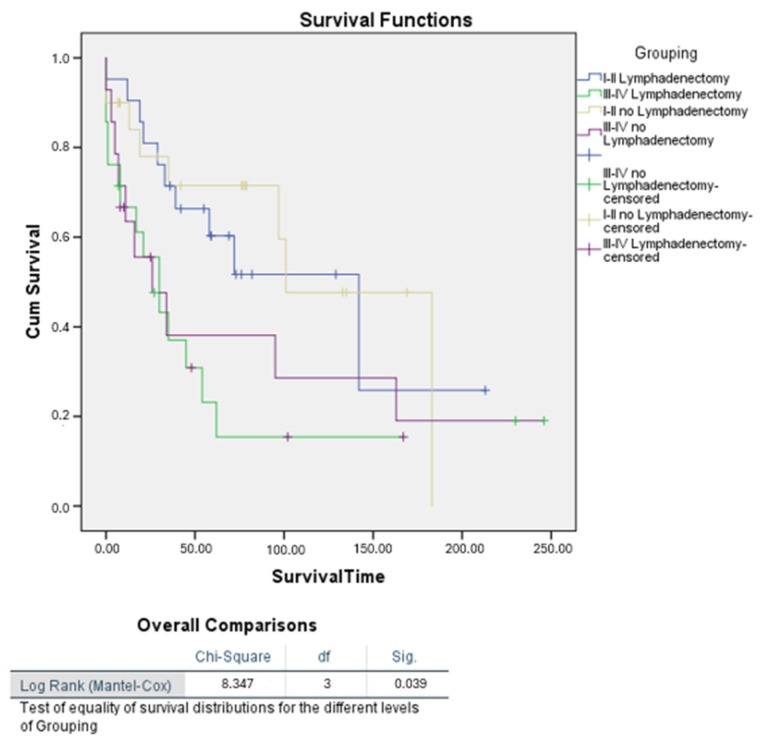
Kaplan–Meier survival curves for patients with uterine carcinosarcoma based on lymphadenectomy status, stratified by FIGO stage.

**Table 1 cancers-16-03905-t001:** Baseline characteristics of the included patients.

Characteristic	N (%)
**Parity**	
Uni-multiparous	60 (75%)
Nulliparous	20 (25%)
**Menopausal Status**	
Postmenopausal	78 (97.5%)
**Comorbidities**	
Hypertension	29 (36.25%)
Diabetes	12 (15%)
History of Breast Cancer	11 (13.75%)
Family History of Cancer	7 (8.52%)
**Initial Symptoms**	
Metrorrhagia	31 (38.75%)
Vaginal Spotting	18 (22.5%)
Asymptomatic	7 (8.75%)
Incidental Diagnosis	2 (2.5%)
Abdominal Pain	19 (23.75%)
Palpable Abdominal Mass	4 (5%)
**Tumour Markers**	
Positive CA-125	15 (18.75%)
Negative CA-125	31 (38.75%)
Unknown CA-125	34 (42.5%)
Elevated CA-15.3/CA-19.9	5 (6.25%)
**ECOG Performance Status**	
ECOG 0	49 (61.25%)
ECOG 1	2 (2.5%)
ECOG 2	2 (2.5%)
Unknown	27 (33.75%)
**Hysteroscopy Findings**	
Polypoid	45 (57%)
Diffusely Irregular	33 (41%)

**Table 2 cancers-16-03905-t002:** Cox proportional hazards regression univariate analyses of age, tumour size, and stage as significant predictors of survival and non-significantly impacting variables, showing hazard ratios (HRs), *p*-values, 95% confidence intervals (CIs), as well as the variable’s type.

Variable	Type	B	SE	Wald	df	*p*-Value	Exp(B) (HR)	95% CI Lower	95% CI Upper
Age	Continuous	0.062	0.020	10.013	1	**0.002**	1.064	1.024	1.105
Tumour Size	Categorical (≤4/>4 cm)	0.767	0.327	5.514	1	**0.019**	2.154	1.135	4.087
Tumour Stage	Categorical (I–II//III–IV)	−0.728	0.349	5.149	1	**0.008**	0.438	0.238	0.809
Histotype	Categorical (heter/homol)	0.186	0.323	0.332	1	0.564	1.205	0.640	2.268
Myometrial Invasion	Categorical (≤50/>50)	0.380	0.354	1.153	1	0.283	1.462	0.731	2.924
Lymphadenectomy	Categorical (Y/N)	−0.226	0.314	0.515	1	0.473	0.798	0.431	1.478

Heter/homol: heterologous/homologous; SE: standard error; Y/N: yes/no. Statistically significant *p*-Value are evidenced.

**Table 3 cancers-16-03905-t003:** Multivariate Cox proportional hazards regression analysis of age, tumour size, and tumour stage as independent predictors of survival, showing hazard ratios (HRs), *p*-values, and 95% confidence intervals (CIs).

Variable	Type	B	SE	Wald	df	*p*-Value	Exp(B) (HR)	95% CI Lower	95% CI Upper
Age	Continuous	0.067	0.021	9.711	1	**0.002**	1.067	1.024	1.111
Tumour Size	Categorical (≤4/>4 cm)	−0.441	0.348	1.601	1	0.206	0.644	0.325	1.274
Tumour Stage	Categorical (I–II//III–IV)	−0.728	0.321	5.149	1	**0.023**	0.483	0.238	0.906

SE: standard error. Statistically significant *p*-Value are evidenced.

**Table 4 cancers-16-03905-t004:** Stepwise multivariate Cox proportional hazards regression analysis results, showing the hazard ratios (HR), *p*-values, and 95% confidence intervals (CI) for age, tumour stage, and other not statistically significant variables (tumour size, histotype, myometrial invasion, lymphadenectomy).

Variable	Type	B	SE	Wald	df	*p*-Value	Exp(B) (HR)	95% CI Lower	95% CI Upper
Age	Continuous	0.045	0.020	4.901	1	**0.027**	1.046	1.005	1.089
Tumour Stage	Categorical (≤4/>4 cm)	−0.763	0.336	5.143	1	**0.023**	0.466	0.241	0.902
Tumour Size	Categorical (I–II//III–IV)	−0.441	0.348	1.601	1	0.206	0.644	0.325	1.274
Histotype	Categorical (heter/homol)	0.186	0.323	0.332	1	0.564	1.205	0.640	2.268
Myometrial Invasion	Categorical (≤50/>50)	0.380	0.354	1.153	1	0.283	1.462	0.731	2.924
Lymphadenectomy	Categorical (Y/N)	−0.226	0.314	0.515	1	0.473	0.798	0.431	1.478

SE: standard error. Statistically significant *p*-Value are evidenced.

## Data Availability

The data presented in this study are available in this article (and Appendix A).

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
