# Peer review of "Uterine Carcinosarcoma (UCS): A Literature Review and Survival Analysis from a Retrospective Cohort Study"

_cancers, 2024, doi:10.3390/cancers16233905_

Round 1
Reviewer 1 Report
Comments and Suggestions for Authors
Structure and clarity of the manuscript
The manuscript is clearly structured and well written. The paper begins with a detailed introduction describing the context and objectives. A comprehensive materials and methods section, a thorough presentation of the findings, and a well-founded discussion follow this. The transitions between the sections are sensible and easy to understand. However, minor changes could further improve clarity, especially in the section on techniques, where some experimental details should be made clearer.
The references included in the manuscript are mostly up-to-date and relevant to the topic, especially when it comes to treatment approaches, molecular profiling and survival analysis of uterine cancer syndrome (UCS), and are from the last five years. There are not too many self-citations in the article, suggesting that the authors have considered a wide range of work on the topic. There is no conspicuous absence of important evidence on the prognosis or management of UCS.
The study design, based on a review of 80 UCS cases from three different universities, is appropriate given the aims of the study. The statistical methods used, such as Cox proportional hazards regression and Kaplan-Meier survival analysis, are reliable and conventional for determining survival predictors. The experimental strategy is valid from a scientific point of view, even though it is an observational and retrospective study, which limits some control of variables. However, a more thorough explanation of the inclusion and exclusion criteria for the patients would improve the replicability of the study.
The methods section provides a reasonable level of detail about the collection of patient data and statistical analysis, but some elements could be expanded to improve reproducibility. For example, more detailed information on how patients were selected and any problems with incomplete data would be helpful. Overall, the approach is sufficiently rigorous for the aims of the study, although future studies would benefit from prospective designs to mitigate the inherent limitations of retrospective cohort studies.
The figures and tables presented are appropriate for the study and support the results well. The Kaplan-Meier survival curves are clear and easy to interpret, with large statistical differences between groups visually apparent. The data are interpreted consistently throughout the publication, with the authors providing coherent explanations for both significant and non-significant results. The limitations of various parts of the study, such as the limited sample size and lack of molecular profiling, are adequately addressed.
The interpretation of the results is coherent and in good agreement with the data. The conclusions of the text are strongly supported by the Cox regression analyzes and the Kaplan-Meier survival curves. Nevertheless, it is important to point out that tumor size had statistical significance in the univariate analysis, although this was not the case in the multivariate analysis. The authors address this fact, taking into account the limitations of the study design. The authors also rightly point out that further research is needed, particularly with regard to the importance of lymphadenectomy in the treatment of UCS.
The conclusions drawn by the authors are supported by the data presented. They rightly emphasize the need for early detection and the influence of tumor size, age and stage on the prognosis of patients with UCS. The authors are careful not to overstate their findings given the limitations of the study, and the commentary skillfully integrates the results of the study into the existing literature.
Explanations on ethics and data availability
The manuscript contains sufficient information on data availability and ethical aspects. All participants gave informed consent and the study was conducted in accordance with the Declaration of Helsinki. The study raises no ethical issues. Furthermore, the study recognizes its limitations as it is a multi-institutional retrospective cohort study, even if the data were not made publicly available.
This paper presents new survival data along with a thorough analysis of uterine carcinoma (UCS) and functions as both an original study and a review. It effectively highlights a gap in our existing knowledge of UCS, particularly in relation to prognostic variables and therapeutic approaches. As UCS is rare and there aren't many prospective studies, this review is nevertheless helpful to the scientific community. The review is well documented and covers most of the recent research, and no obvious gaps in knowledge have been overlooked.
The reproducibility of the study could be improved by a more thorough explanation of patient inclusion criteria and any confounding factors.
It would be helpful to clarify why some factors (such as tumor size) become less important in multivariate models.
Overall, the paper is well presented, scientifically sound and relevant to the field. It contributes to the state of knowledge, points directions for further investigation and provides insightful information on prognosis and treatment approaches for UCS. The methodological description and data transparency could both be slightly improved to enhance the quality of this work.
Author Response
We would like to express our sincere gratitude to the Reviewer for taking the time to provide valuable feedback on our manuscript. We appreciate your thoughtful comments, which have guided us in further refining our study. We are pleased to address each point raised and have carefully revised the manuscript to enhance its clarity, methodological rigor, and overall contribution to the understanding of uterine UCS.
We hereby respond to each comment individually, citing the changes made in the text
Comment 1: "The manuscript is clearly structured and well written. The paper begins with a detailed introduction describing the context and objectives. A comprehensive materials and methods section, a thorough presentation of the findings, and a well-founded discussion follow this. The transitions between the sections are sensible and easy to understand. However, minor changes could further improve clarity, especially in the section on techniques, where some experimental details should be made clearer."
Response 1: We appreciate the reviewer’s positive feedback on the structure and clarity of our manuscript. In response to the request for further clarity, we have expanded the methods section to provide additional detail on patient selection criteria and data completeness.
Change made - In the Materials and Methods section, we included a more thorough explanation of the selection process, detailing the specific inclusion and exclusion criteria, as follows: We included patients with histologically confirmed uterine carcinosarcoma (UCS) at any stage of the disease, who received both primary treatment and follow-up care at one of the three participating hospitals. Eligible patients had to be aged 18 years or older, as the study focused on adult populations typically affected by UCS. Only patients who had comprehensive medical records with information relevant to the study’s objectives, such as clinical symptoms, diagnostic and surgical details, tumor characteristics, and follow-up data, were considered in this study. Patients were excluded if they had incomplete medical records or insufficient follow-up data, as these could limit accurate assessments of survival outcomes. For ethical reasons, patients under 18, pregnant women, and those with psychological conditions that would preclude informed consent were excluded. Additionally, patients with significant comorbidities that could confound survival data, such as other advanced malignancies, were also excluded to minimize bias in interpreting UCS-specific prognostic outcomes
Comment 2: "The study design, based on a review of 80 UCS cases from three different universities, is appropriate given the aims of the study. The statistical methods used, such as Cox proportional hazards regression and Kaplan-Meier survival analysis, are reliable and conventional for determining survival predictors. The experimental strategy is valid from a scientific point of view, even though it is an observational and retrospective study, which limits some control of variables. However, a more thorough explanation of the inclusion and exclusion criteria for the patients would improve the replicability of the study."
Response 2: We agree with the reviewer that a detailed explanation of inclusion and exclusion criteria strengthens replicability and provides clarity. Therefore, we added specific criteria defining both the target population and the study sample.
Change made – see response 1
Comment 3: "The methods section provides a reasonable level of detail about the collection of patient data and statistical analysis, but some elements could be expanded to improve reproducibility. For example, more detailed information on how patients were selected and any problems with incomplete data would be helpful."
Response 3: We have revised the methods to address issues related to incomplete data and potential selection bias, including details on how missing data were managed.
Change made: The Materials and Methods section now contains: To ensure reliability, patients with follow-up periods shorter than six months were excluded unless death occurred, as limited follow-up could lead to incomplete survival data. Additionally, to address missing data concerns, two independent researchers reviewed all cases for eligibility and resolved any disagreements through consensus.
Comment 4: "The interpretation of the results is coherent and in good agreement with the data. The conclusions of the text are strongly supported by the Cox regression analyses and the Kaplan-Meier survival curves. Nevertheless, it is important to point out that tumor size had statistical significance in the univariate analysis, although this was not the case in the multivariate analysis. The authors address this fact, taking into account the limitations of the study design."
Response 4: We acknowledge the reviewer’s observation regarding tumor size's statistical significance in univariate but not multivariate analyses. We have clarified this more in the discussion section.
Change Made: The Discussion section now includes: Although tumour size did not retain its significance in the multivariate model, likely due to the confounding influence of advanced-stage disease […]
Comment 5: "The reproducibility of the study could be improved by a more thorough explanation of patient inclusion criteria and any confounding factors."
Response 5: We agree that accounting for confounding variables enhances study robustness. In response, we detailed potential confounders in our analysis.
Change made: The Materials and Methods section was updated as follows: Potential confounding factors, such as age, tumor stage, and comorbidities, were controlled for in the multivariate Cox proportional hazards regression. This adjustment aimed to minimize bias and improve the reliability of our findings regarding UCS-specific survival predictors.
Comment 6: "It would be helpful to clarify why some factors (such as tumor size) become less important in multivariate models."
Response 6: We recognize that tumor size’s diminished impact in the multivariate model may need further clarification. In response, we added an explanation in the discussion regarding interactions between tumor size and stage. See also response 4.
Change Made: The Discussion section now reads: The reduced significance of tumor size in multivariate analysis suggests that advanced stage may subsume the impact of tumor size on survival, as larger tumors are often associated with later-stage disease. This pattern indicates that stage may be a more comprehensive marker of disease progression in UCS.
We hope these revisions address the reviewer’s suggestions, providing a more detailed, transparent account of our methods, criteria, and analytical interpretations. Thank you again for the constructive feedback, which has greatly enhanced the clarity and rigor of our manuscript.
Sincerely,
Dr M.F.P. Maiorano
Reviewer 2 Report
Comments and Suggestions for Authors
I read the manuscript with great interest. It is a very interesting paper, with a broad focus on a very complex and aggressive pathology: uterine carcinosarcoma.
The discussion of the pathology is very well conducted, with a broad excursus on this terrible problem.
The objective of the study has been largely respected: to identify key predictors of survival and explore their potential associations, ultimately contributing to improve management strategies for this rare and aggressive malignancy.
The number of tables is correct, as are the graphs on logistic regression.
I think there is nothing else to add to this excellent research.
I congratulate the authors.
Author Response
Comment 1: I read the manuscript with great interest. It is a very interesting paper, with a broad focus on a very complex and aggressive pathology: uterine carcinosarcoma. The discussion of the pathology is very well conducted, with a broad excursus on this terrible problem. The objective of the study has been largely respected: to identify key predictors of survival and explore their potential associations, ultimately contributing to improve management strategies for this rare and aggressive malignancy. The number of tables is correct, as are the graphs on logistic regression. I think there is nothing else to add to this excellent research.I congratulate the authors.
Response 1:
Dear Reviewer,
Thank you for your thoughtful and encouraging feedback on our manuscript. We are grateful for your kind words and are pleased to know that you found the study engaging and valuable.
We are especially glad that the discussion, study objectives, and focus on uterine carcinosarcoma’s complexities resonated with you. Your comments on the manuscript’s structure, including the number of tables and the logistic regression graphs, are highly appreciated, as we aimed to provide a clear and comprehensive overview of our findings.
Thank you once again for your supportive review and for acknowledging the efforts we put into this research.
Sincerely,
Dr M.F.P. Maiorano
Reviewer 3 Report
Comments and Suggestions for Authors
In the manuscript entitled “Uterine Carcinosarcoma (UCS): Literature Review and Survival Analysis from a Retrospective Cohort Study” the authors present data from retrospective study of 80 patients of UCS. They found that tumor stage and age are significant independent predictors of survival in UCS and lymphadenectomy was not independently associated with improved survival.
The findings are of interest, there are a few corrections that need to be made in order to be published.
1) Introduction is too long and I suggest shortening it. There is some confusion with leiomyosarcoma, especially in the epidemiology section.
2) In Material and Methods phase, the authors should describe the details of histological characteristics. Does it include only tumor size and characterizing sarcoma dominance pattern? Is the tumor size the maximum diameter?
3) I suggest that a more detailed table of clinical characteristics.
4) This study included cases over a 30-year period. The FIGO classification has changed during that time, did you have a reclassification?
5) In Cox proportional hazards regression univariate analyses, is age a continuous variable? Is the tumor size divided into two groups, less than 4 cm and more than 4 cm? Or is it a continuous variable? There is few information in the text, but I suggest that you also provide more details in Table2
6) Please edit Kaplan-Meier's graph to make it a little more readable. Kaplan-Meier survival analysis stratified by tumor stage only should be included. Figure 6 is considered unnecessary since it is just a combination of Figure 7 and Figure 8.
Author Response
Thanks to the reviewer for his/her thorough review and insightful suggestions. We greatly appreciate the time and consideration dedicated to improving the clarity and rigor of our manuscript. In response to each of the comments, we have made several modifications, as detailed below, to address concerns and enhance the study's clarity, readability, and scientific rigor.
Please, find our point-by-point responses and the corresponding changes made to the manuscript.
Comment 1: Introduction is too long and I suggest shortening it. There is some confusion with leiomyosarcoma, especially in the epidemiology section.
Response 1: We have refined the epidemiology section to eliminate any potential confusion with leiomyosarcoma. Specifically, we clarified that UCS accounts for approximately 2-5% of all uterine malignancies, with an estimated incidence of 2-5 per 100,000 women in the United States, and noted its association with high mortality rates due to its aggressive nature.
Change made: "Due to the rarity of UCS and the complexity in their diagnosis, epidemiological data are limited. In the United States, UCS constitutes approximately 2-5% of all uterine malignancies, with an estimated incidence of 2-5 per 100,000 women per year. Their prognosis remains poor, with a 5-year survival rate between 18% and 47%, accounting for 16.4% of deaths caused by uterine malignancies despite their rarity”. We also carefully shortened the introduction while maintaining its purpose as a combined review of the UCS state of the art, given the limited prospective research available on this rare and aggressive malignancy. Our intent is to provide readers with a comprehensive context for understanding the study's objectives, as UCS is an uncommon and complex pathology that benefits from a thorough overview of existing knowledge.
Comment 2: In Material and Methods phase, the authors should describe the details of histological characteristics. Does it include only tumor size and characterizing sarcoma dominance pattern? Is the tumor size the maximum diameter?
Response 2: We have expanded the description of histological characteristics, detailing that the tumor size represents the maximum diameter of the tumor, as you inquired. Additionally, we clarified that histological features included the assessment of both sarcomatous and carcinomatous components, including the evaluation of sarcomatous dominance (where the sarcomatous component exceeds 50% of the tumor), as well as the differentiation types within these components.
Change made: "Tumor histology was assessed to include the maximum diameter (tumor size) and the presence and proportion of sarcomatous and carcinomatous components, noting whether the sarcomatous component was homologous or heterologous and if sarcomatous dominance (greater than 50% of the tumor) was observed. The specific differentiation patterns (e.g., rhabdomyosarcoma, chondrosarcoma) within the sarcomatous component, as well as high-grade serous, endometrioid, or undifferentiated forms within the epithelial component, were documented."
Comment 3: I suggest that a more detailed table of clinical characteristics.
Response 3: Thank you for your suggestion to provide a more detailed table of clinical characteristics. We have expanded Table 1 to include additional baseline details.
Change made: The updated Table 1 now includes: Initial symptoms, CA-125 status, ECOG performance scores, and hysteroscopy findings”
Comment 4: This study included cases over a 30-year period. The FIGO classification has changed during that time, did you have a reclassification?
Response 4: We used FIGO 2009 staging classification, and reclassification were made if previous or different staging systems were used.
Change made: We changed the Methods section accordingly: “based on the FIGO 2009 system for endometrial carcinoma [45]: restaging were made accordingly if previous or different staging systems were used”
Comment 5: In Cox proportional hazards regression univariate analyses, is age a continuous variable? Is the tumor size divided into two groups, less than 4 cm and more than 4 cm? Or is it a continuous variable? There is few information in the text, but I suggest that you also provide more details in Table2
Response 5: we specified the nature of the variables in the methods section. We also added details regarding variable’s types in Table 2. We did not add the same in all the tables to make them more readable.
Change made: We added the folllwing in the method section: “In the Cox regression analyses, age was treated as a continuous variable, while tumour size was categorized into two groups: ≤4 cm and >4 cm, based on the maximum diameter of the tumour. Tumour stage (I-II vs III-IV), myometrial invasion (≤50% and >50%), histotype (homologous and heterologous) and lymphadenectomy (performed/not performed) were considered categorical variables in the analyses, as specified in Table 2”. We modified Table 2 accordingly.
Comment 6: Please edit Kaplan-Meier's graph to make it a little more readable. Kaplan-Meier survival analysis stratified by tumor stage only should be included. Figure 6 is considered unnecessary since it is just a combination of Figure 7 and Figure 8.
Response 6: We added a new Figure 1 - Kaplan-Meier survival curves for patients with uterine carcinosarcoma stratified by tumor stage only, writing a new section in the results and discussion paragraphs. We deleted Figure 6 as suggested.
Change made: Added the following in the results section: “Kaplan-Meier survival analysis by tumor stage revealed a statistically significant difference in survival between patients with early-stage (Stage I-II) and advanced-stage (Stage III-IV) uterine carcinosarcoma (log-rank p = 0.027). (Figure 1)”
Added in the discussion section: “The Kaplan-Meier survival analysis stratified by FIGO stage revealed significant survival differences, with patients in Stage I-II demonstrating markedly better survival compared to those in Stage III-IV (log-rank p = 0.027). Specifically, the median survival was estimated at 101 months for Stage I-II, while it was only 30 months for Stage III-IV, emphasizing tumor stage as a pivotal determinant of survival in UCS. This finding is consistent with our multivariate Cox regression, which identified tumor stage as the most significant independent predictor of survival, where advanced stages (Stage III-IV) were associated with a substantially higher risk of mortality (HR = 0.438, p = 0.008). These observations reinforce established knowledge that early-stage diagnosis is associated with improved survival, underscoring the critical need for early detection strategies in UCS management [5, 49, 59]”
Sincerely,
Dr. M.F.P. Maiorano
Round 2
Reviewer 3 Report
Comments and Suggestions for Authors
This second version of the paper is improvement, but I have found a few issues that, once addressed, will improve the manuscript.
1) As I pointed out in my initial review, “Introduction” is too long and I suggest shorting it. Instructions for authors in “Cancers” (https://www.mdpi.com/journal/cancers/instructions#preparation) include the following.
“The introduction should briefly place the study in a broad context and highlight why it is important.”
2) I am sorry I did not point out in detail, but there is no unity between the FIGURE and the TABLE.
Please add details regarding variable in Table3 and 4.
Please add a figure regarding homologous v.s. heterologous.
Please change the order of Figures 3 and 4.
Please add a figure regarding lymphadenectomy v.s. no lymphadenectomy.
Please put Figures 7 and 8 together as you would any other.
Author Response
We want to thank the reviewer for these valuable comments on our article and the clarification on his/her first round of review. We updated the manuscript as follows
Comment 1: As I pointed out in my initial review, “Introduction” is too long and I suggest shorting it. Instructions for authors in “Cancers” (https://www.mdpi.com/journal/cancers/instructions#preparation) include the following.
“The introduction should briefly place the study in a broad context and highlight why it is important.”
Response 1: we significantly cut the introduction part as suggested
Comment 2: I am sorry I did not point out in detail, but there is no unity between the FIGURE and the TABLE. Please add details regarding variable in Table3 and 4. Please add a figure regarding homologous v.s. heterologous. Please change the order of Figures 3 and 4. Please add a figure regarding lymphadenectomy v.s. no lymphadenectomy. Please put Figures 7 and 8 together as you would any other.
Response 2: we added the variables' detail in the Table 3 and 4. We added a Figure regarding homologous vs heterologous. We changed previous Figure 3-4 order. We added the figure regarding lymphadenectomy. Finally we added the image regarding lymphadenectomy for all stages, putting Figure 7 and 8 together.
We hope that the revised manuscript meets the reviewer's need for more clarity and concision.
Thank you for the time and consideration.
Sincerely,
Dr. M.F.P. Maiorano
Round 3
Reviewer 3 Report
Comments and Suggestions for Authors
No specific comments.